# An Examination of Demographic and Psychosocial Factors, Barriers to Healthy Eating, and Diet Quality Among African American Adults

**DOI:** 10.3390/nu11030519

**Published:** 2019-02-28

**Authors:** Ingrid K. Richards Adams, Wilson Figueroa, Irene Hatsu, James B. Odei, Mercedes Sotos-Prieto, Suzanne Leson, Jared Huling, Joshua J. Joseph

**Affiliations:** 1College of Food, Agricultural, and Environmental Sciences, Ohio State University Extension, Columbus, OH 43201, USA; Figueroa.123@osu.edu (W.F.); hatsu.1@osu.edu (I.H.); 2Medical Dietetics, School of Health and Rehabilitation Sciences, The Ohio State University College of Medicine, 453 W. 10th Ave., Atwell Hall 306 D, Columbus, OH 43210, USA; Suzanne.Leson@osumc.edu; 3Department of Human Sciences, College of Education and Human Ecology, The Ohio State University, 341 Campbell Hall, 1787 Neil Ave, Columbus, OH 43210, USA; 4Division of Biostatistics, College of Public Health, 248 Cunz Hall, 1841 Neil Avenue, Columbus, OH 43210, USA; odei.3@osu.edu; 5Food and Nutrition Sciences, Ohio University College of Health Sciences and Professions, Grover Center E189, Athens, OH 45701, USA; sotospri@ohio.edu; 6Department of Statistics, Ohio State University, 329 Cockins Hall, 1958 Neil Ave, Columbus, OH 43210, USA; huling.7@osu.edu; 7Division of Endocrinology, Diabetes and Metabolism, The Ohio State University Wexner Medical Center, 566 McCampbell Hall, 1581 Dodd Dr., Columbus, OH 43210, USA; joseph.117@osu.edu

**Keywords:** African Americans adults, healthy eating index, barriers to healthy eating

## Abstract

A healthy diet is associated with lower risk of chronic disease. African Americans generally have poor diet quality and experience a higher burden of many chronic diseases. We examined the associations of demographic and psychosocial factors and barriers to diet quality among African American adults. This cross-sectional study included 100 African American adults in a southeastern metropolitan area. Psychosocial factors (social support, self-efficacy), and barriers to healthy eating were assessed with validated measures. Diet quality was assessed using the Healthy Eating Index (HEI-2010). Nested linear regressions were used to examine the association between the variables of interest and HEI scores. Participants reported having social support (M (mean) = 2.0, SD (standard deviation) = 0.6, range 0–3), high levels of self-efficacy (M = 3.1, SD = 0.7, range 1–4), and low barriers (M = 1.4, SD = 0.6, range 0–4) to engage in healthy eating but total mean HEI scores needed improvement (M = 54.8, SD = 10.9, range 27.1–70.0). Participants consumed significantly higher empty calories and lower whole fruits, dairy, and total protein foods than the national average. Barriers to healthy eating (b = −12.13, *p* = 0.01) and the interaction between age and barriers (b = 0.25, *p* = 0.02) were most strongly associated with lower HEI scores. Younger African Americans with the highest barriers to healthy eating had the lowest HEI scores. Culturally appropriate interventions targeting empty calories, barriers to healthy eating, and knowledge of the Dietary Guidelines for Americans are needed for African Americans.

## 1. Introduction

Amidst reports of health improvements, African Americans living in the United States continue to be disproportionately impacted by chronic disease resulting in higher death rates compared to non-Hispanic whites [1,2]. African Americans are 1.4 times as likely to be obese, 1.5 times more likely to have hypertension, and are more than twice as likely to have diabetes or a stroke than their non-Hispanic white counterparts [3,4,5,6,7]. This disparity leads to greater reduction in income (due to lost productivity), and higher cost for care in terms of responsibilities and medical expenses. Chronic disease-related expenses cost the nation $3.3 trillion annually and are expected to increase to $42 trillion by the year 2030 [8,9,10]. Although chronic diseases are among the most debilitating and costly conditions, they are also among the most preventable.

Poor diet quality is a leading risk factor for chronic diseases and consumption of high quality diets have been associated with significantly lower risk in chronic disease and all-cause mortality [11,12,13,14,15]. A high diet quality reflects an eating pattern that includes a variety of vegetables, fruits, whole grains, protein, low fat dairy, and healthy oils, and limited intake of saturated and trans fats, added sugars, and sodium [16]. A recent systematic review and meta-analysis of prospective studies showed an association between high diet quality and lower risk of cancer and cardiovascular disease incidence and mortality, type 2 diabetes, and neurodegenerative diseases. In addition, high diet quality among cancer survivors was associated with a significant decrease in all-cause mortality [14]. The improvement in adherence to diet quality was also shown to lower both short and long-term cardiovascular risks [17,18].

Despite these strong associations, few studies have examined diet quality among African American adults [19,20,21]. Instead, the majority of studies have focused on dietary components rather than overall diet quality. Results from these studies indicate that indeed African Americans have lower intakes of dietary fiber and vegetables, whole grains, and dairy, and higher intakes of fat, sodium, and sugar-sweetened beverages [22,23,24,25,26,27]. Although this information adds to the extant literature, it is limiting in that it neither provides a holistic picture of overall diet quality nor of how the diets of African Americans are aligned with the recommendations of the Dietary Guidelines for Americans (DGA). This gap in the literature precludes the use of such empirical evidence by nutrition educators and policy makers in targeted interventions, while also masking diet-related disparities among African Americans. Boggs et al. noted that due to differences in disease incidence and comorbidities, genetic predisposition, and unique modifying factors between ethnic groups, a focus on diet quality among African American is paramount [20].

Diet-related disparities are complex, multifaceted, and associated with demographic (age, gender, education, income) and psychosocial (self-efficacy, social support) factors and influences, as well as barriers to healthy eating [26]. Past research has shown a positive linear association between socioeconomic status (SES) and diet quality [28,29]. This relationship is evident even in international populations [30]. Several studies have also shown that having social support (or positive social influences and encouragement from others to engage in healthy eating) was associated with higher diet quality [31,32,33,34]. Similarly, self-efficacy has been shown to predict diet quality in different target populations [35,36,37,38]. Barriers to healthy eating such as cost and finances, availability and convenience, absence of fruits and vegetables in the home, and lack of knowledge regarding healthy foods have also been associated with the diet quality of African Americans [39,40].

Although several studies have examined individuals factors associated with diet-related disparities, few have collectively examined the demographic and psychosocial factors, and barriers to healthy eating with diet quality among African Americans. The majority of studies consisted of mixed populations with results not disaggregated by race, thus making it difficult determine the true effect of these factors on the diet quality of African Americans. The goal of this study was to collectively examine the association of social support, self-efficacy, barriers to healthy eating, and demographic variables with the diet quality of African American adults in order to inform, and tailor future interventions and education and policy efforts.

## 2. Materials and Methods

This study is a secondary analysis of data from a parent study conducted in 2008–2009 that employed a cross sectional design to examine perceptions of healthy eating and physical activity among African American adults in Lexington, Kentucky. Individuals were recruited through African American churches, newspaper advertisements, bulletin boards, and snowballing techniques, where individuals who were informed of the study contacted other individuals in their social network. Individuals were included if they identified as African American, were between the ages of 18 and 74, and had resided in the greater metropolitan area for at least 1 year. Participants were compensated $25 for their time. The procedures were approved by the Institutional Review Board of The University of Kentucky and all participants gave written informed consent. All subjects gave their informed consent for inclusion before they participated in the study. The study was conducted in accordance with the Declaration of Helsinki, and the protocol was approved by the Ethics Committee of the University of Kentucky (09-0200-P4S).

### 2.1. Measures

The REACH 2010 St. Louis Healthy Heart Survey (instrument): an 89 item closed-ended multiple choice survey (some of which included skip logic) was used to collect information on demographic (age, sex, income and education) variables, psychosocial factors (self-efficacy and social support), and barriers to healthy eating [41]. The survey was based on questions from the Behavioral Risk Factor Surveillance System [42,43,44] and the Missouri Cardiovascular Disease Telephone Survey (both of which have been tested for reliability and validity) [45,46,47].

#### Constructs 

Barriers to healthy eating: Questions included factors that prevented individuals from engaging in healthy eating. The nine factors were as follows: (1) “Others discourage me”, (2) “I don’t have the time to prepare healthy foods”, (3) “I am too tired to prepare healthy food”, (4) “Fast food is more convenient”, (5) “Healthy food costs too much”, (6) “I don’t like the taste of healthy food”, (7) “I don’t like to eat healthy food”, (8) “I don’t have access to healthy food”, and (9) “I don’t know how to prepare low fat or healthy food”. Nine barrier items were assessed and scored on a scale of 0–4 (0 = “never”, 1 = “rarely”, 2 = “sometimes”, 3 = “often”, 4 = “very often”), with higher scores indicating greater barriers. A composite score was calculated by averaging the nine items. Reliability of the barriers to healthy eating items used in this study was *α* = 0.70.

Self-efficacy for healthy eating: Self-efficacy is confidence or belief in one’s own ability to engage in a desired behavior, here healthy eating [48]. The six questions related to self-efficacy towards healthy eating were: (1) “Could you eat low fat or healthy food when at a restaurant?” (2) “Could you eat low fat or healthy food when you are at a family or social gathering?” (3) “Could you avoid high fat snack foods when you are in a hurry and don’t have time to prepare a healthy meal?” (4) “Could you eat five fruit and vegetables every day?” (5) “Could you request low fat or healthy food options from a grocery store?” and (6) “Could you request low fat or healthy food options when at a restaurant?” Six self-efficacy items were assessed and scored on a scale of 1–4 (1 = “I’m sure I couldn’t do it”, 2 = “maybe I couldn’t do it”, 3 = “maybe I could do it”, 4 = “I’m sure I could do it”) with higher scores indicating greater self-efficacy. A composite score was calculated by averaging the six items. Reliability for the self-efficacy items was *α* = 0.85.

Social support to healthy eating: This refers to positive influences and encouragement from others to engage in healthy eating. Questions related to social support included, “Your friends encourage you to eat healthy”, “You have at least one friend who would commit to eat healthy with you”, and “Relatives encourage you to eat healthy.” The five social support items were assessed and scored from 0–3 (0 = “strongly disagree”, 1 = “disagree”, 2 = “agree”, 3 = “strongly agree”) with higher scores indicating greater social support towards healthy eating. A composite score was calculated by averaging the five items. Reliability for the social support to healthy eating items used in this study was *α* = 0.90.

Diet Quality: The full length Block Food Frequency Questionnaire 2005 (FFQ) was used to determine the nutrient and dietary intake of participants [49,50]. The Block FFQ is a validated measure with a food and beverage list that includes 127 items, plus supplementary questions to allow for the adjustment of fat, protein, carbohydrate, sugar, and whole grain content. The questionnaire ascertains the frequency with which each food or beverage was usually consumed, and offered nine continuous responses ranging from “never” to “every day” for most foods. In addition, portion size is asked for each food/beverage item, with pictures provided to improve accuracy of estimation. The FFQs were self-administered and took approximately 30 min to complete. The analyzed food and nutrient intake data from the FFQ were then used to calculate diet quality based on the Healthy Eating Index 2010 (HEI) scores. The HEI is a validated diet quality measure that assesses how closely an individual’s dietary pattern conforms to the recommendations of the Dietary Guidelines for Americans [16]. It includes 12 components, nine of which (total fruit, whole fruit, total vegetables, greens and beans, whole grains, dairy, total protein foods, seafood and plant proteins, fatty acids) are recommended to be consumed in adequate amounts, and three of which (refined grains, sodium and empty calories) are to be consumed in moderation [51]. For all components, higher scores demonstrate compliance with dietary guidelines, i.e., higher scores of the “adequacy” components correspond to higher intake of theses dietary components, while higher scores on “moderation” components are indicative of lower intake of these components [51]. The standards for determining adequacy and moderation scoring standards are described in details elsewhere [52]. Briefly, it includes: (1) identifying the group of foods under consideration (e.g., the total amount of foods consumed in a day), (2) linking these foods to relevant databases to determine the amounts of each relevant dietary constituents in these foods, and (3) deriving pertinent ratios to compare to relevant standards for scoring [53]. A total HEI score (range 0–100) is calculated from the 12 component scores with higher HEI scores indicative of a better diet quality and greater adherence to dietary guidelines [51]. In the current study, both total HEI score and subcomponents were examined. It should be noted that national average derived from the National Health and Nutrition Survey also used the HEI 2010 and Block FFQ-2005.

### 2.2. Data Analysis

Data were examined for variable distribution and outliers with statistical and graphical methods. No significant outliers were found, however five participants were missing data on either the exposure or outcome and therefore the final analytic sample included 100 participant. Data were examined to characterize the sample and descriptive statistics were used to summarize the variables of interest. Nested linear regressions were used to assess the association between the variables of interest (demographic variables, self-efficacy, social support, and barriers to healthy eating) and HEI scores. In model 1, we examined demographic variables, model 2 added self-efficacy to healthy eating, model 3 added social support to healthy eating model 4 added barriers to healthy eating and the interaction between barriers and age. The final model explained 20% of the variance in HEI scores. One-sample t-tests were conducted to examine differences in subcomponents scores of the HEI between sample participants and the national average. Pearson correlations were used to identify and evaluate collinearity between these variables. Statistical significance was established at *p <* 0.05. Analyses were conducted using R version 3.5.1 (R Foundation for Statistical Computing, Vienna, Austria) [54].

## 3. Results

### 3.1. Characteristics of Participants

The majority of the participants were female (70.0%) and had an income at or below $35,000. Approximately half of the participants (49%) were between the ages of 18 and 35 years, (mean 36.2 years, SD = 15.8) and had some college education. (68.0%). Over half of the participants (58%) had diets that needed improvements (HEI scores 51.93–76.75) and 42% had poor diet quality (HEI scores < 51). No participant had a diet rated in the “good” category (HEI > 80). The total mean HEI score was 54.8 (M = 54.8, SD = 10.9, range = 30.09–76.75) [11]. See Table 1.

### 3.2. Subcomponents of the Healthy Eating Index

Compared to national averages, participants had significantly poorer intakes of the following HEI components: whole fruits (2.8 (1.5) vs. 3.5 (0.2), *p <* 0.001), dairy (4.1 (1.9) vs. 5.7 (0.2), *p <* 0.001), total protein (4.5 (0.8) vs. 5.0 (0.0), *p <* 0.001), and higher intakes of greens and beans, total fruit, whole grains, and fatty acids. Additionally, participants also had significantly higher intakes of empty calories (8.5 (4.4) vs. 11.0 (0.2), *p <* 0.001). Overall, however, there was no significant difference in mean HEI scores between our participants and the national average (*p* = 0.52). See Table 2.

### 3.3. Barriers to Healthy Eating

Nine items were used to assess barriers to healthy eating. Overall, barriers to healthy eating were low in the present sample of participants (M =1.4, SD = 0.6, range 0–4). The most frequently reported barriers was fast food being more convenient (52%) followed by the cost (29%) and lack of time (25%) to prepare healthy food. See Table 3.

### 3.4. Self-Efficacy towards Healthy Eating

Overall, participants’ total self-efficacy towards healthy eating was high (mean = 3.1, SD = 0.7). Highest levels of self-efficacy (“I am sure I could”) were reported for the items “Could you eat low fat or healthy food when at a restaurant?” and “Could you request low fat or healthy food options when at a restaurant?” (48.0% and 50.0%, respectively). The two items that participants reported the lowest self-efficacy in were avoiding high fat snacks when in a hurry (28%) and eating low fat or healthy foods when at a family or social gathering (34%). See Table 4.

### 3.5. Social Support for Heathy Eating

On average, most participants reported that they had social support for health eating (mean = 2.0, SD = 0.6, range 0–3). The majority of the participants agreed or strongly agreed that if they had someone like a friend or a family member to eat healthier with, the chances of eating healthier more often would be greater (91%; Table 5).

### 3.6. Bivariate Correlations

Bivariate correlations showed social support and self-efficacy to engage in healthy eating and education to be positively associated with HEI scores. In other words, those who had higher education and reported greater social support and self-efficacy to engage in healthy eating also had higher HEI scores or diet quality barriers towards healthy eating were negatively associated with HEI scores, those with the highest barriers had the lowest HEI scores. No other variables (i.e., age, income, and sex) were significantly associated with HEI scores. See Table 6.

### 3.7. Linear Regressions

In the presence of all other primary variables of interest (i.e., self-efficacy and social support to healthy eating), barriers to healthy eating were most strongly associated with HEI scores, (b = −12.13, *t* = 2.73, *p* = 0.01). For every one-point increase in the barriers to healthy eating composite score there was a 12.13 point decrease in HEI scores. The interaction between barriers to healthy eating and age predicting HEI scores was examined to see if age potentially moderated the association between barriers to healthy eating and HEI scores. There was a significant interaction between age and barriers to healthy eating (b *=* 0.25, *p* = 0.01), predicting HEI scores. For example, the results of the interaction in Table 7 imply that for those with no barriers to healthy eating, a decrease in age by 1 year was associated with an increase in average HEI scores of 0.26. For those with one barrier to healthy eating age had no impact on HEI scores, and for those with two barriers to healthy eating, a decrease in age by 1 year was associated with a decrease in average HEI scores of 0.25. See Table 7 for the results of model 4.

## 4. Discussion

African American adults have the highest rates of chronic disease, morbidity, and mortality, and generally have the lowest diet quality of all racial groups in the United States [1,55]. This study collectively examined demographic (age, gender, education, income) and psychosocial (self-efficacy, social support) factors, barriers to healthy eating, and diet quality among African American adults. Participants had a mean HEI score that was similar to the national average (“needs improvement”) but none had an HEI score in the “good” category. Among the variables examined, barriers to healthy eating was the strongest predictor of diet quality. There was also an interaction between age and barriers so that younger African Americans with the highest barriers had the lowest diet quality. Despite participants’ reports of having social support, high self-efficacy, and low barriers to healthy eating, diet quality was still in the category of needing improvement.

The mean HEI score of participants in this study was similar to the mean HEI score of the United States population. This result differed from previous findings that suggested African Americans had comparatively poorer diet quality [56,57]. One reason this sample had a diet quality similar to that of the national average, may be due to their high levels of education. The majority of participants (72%) had either some college or a college degree. Hiza and colleagues examined the diet quality of Americans based on age, sex, race/ethnicity, income, and education levels. They found adults with higher levels of education had higher diet quality scores than adults with lower levels of education [55]. Additionally, Wilcox et al. examined diet quality among predominantly African American participants and their results also showed poorer diet quality among participants with lower education [28]. Level of education may have also been associated with the intake of dietary components. For example, Participants consumed a mix of foods and beverages that were significantly lower in refined grains, whole fruit, dairy and protein, and higher in whole grains, greens and beans, total fruit, and fatty acids than the national average. Similar dietary patterns have been observed in populations with higher education. A study by Rehm and colleagues using NHANES data showed that over time (from 1999 to 2012), those with higher education consumed higher amounts of whole grains, whole fruit, and fruit juice compared to those with lower education. Given that the majority of our participants had either some college or a college degree, this could have explained the similarities in mean HEI scores and dietary components with the national sample [57].

Previous research has also shown that African Americans consume fewer servings of milk and dairy products and therefore have lower calcium intake [58,59]. The role of calcium in the reduction of risk for obesity, hypertension, cardiovascular disease, insulin resistance syndrome, and some types of cancers is well established in the literature [59,60,61,62,63]. African Americans and other minority populations tend to be lactose-intolerant, therefore making it difficult to consume dairy products [64]. Bronner et al. suggested the importance of nutrition education interventions that address the cultural, community, and environmental barriers associated with reduced dairy intake among African Americans [61]. Participants consumed less whole fruit but more total fruit. Past research has also shown that African Americans tend to consume less whole fruit compared to other racial groups [55]. It is important to develop strategies to increase the consumption of whole fruit among African Americans as some studies show that 100% fruit juice accounts for approximately one-third of total fruit, which are generally lower in fiber and are source of liquid calories [65,66].

Participants consumed significantly higher amounts of empty calories (subgroups include added sugar, solid fats, and alcohol) than the national average. Major sources of solid fats in the American diet include pizza, grain-based desserts, regular cheese and fatty meats while approximately three-quarters of added sugars originate from sugar-sweetened beverages, snacks, and sweets. Previous research showed African Americans have higher intake of sugar-sweetened beverages, energy from salty snacks, and dessert and sweets [57,67,68]. Kant and Graubard mentioned that non-Hispanic blacks were the only ethnic group that did not show a decrease in per capita energy in the last decade [68]. Interestingly enough, although participants in this study reported high self-efficacy to engage in healthy eating overall, they also reported having very low self-efficacy with regard to avoiding high fat snacks when in a hurry. This result showed some alignment with the high HEI scores observed for empty calories among participants. This study alludes to the importance of having information on dietary quality as well as the dietary components of particular groups before planning nutrition interventions. Future interventions should target the reduction in sugar-sweetened beverages and all sources of empty calories among African Americans as a means of improving diet quality.

A significant predictor of diet quality in the current study was barriers to healthy eating. This finding was interesting as, on average, participants reported a low number of barriers. Although barriers to healthy eating was associated with poor diet quality among the entire sample, this association was stronger for younger African Americans with the greatest number of barriers. Previous research also indicated that young African American adults have the worst diet quality [69,70]. The three most reported barriers to healthy eating in the current study were lack of time to prepare healthy food, cost, and the convenience of fast food restaurants. Similar barriers to healthy eating have been reported in previous research among African American adults [71,72]. Studies have shown that reducing barriers to healthy eating, for example making time to prepare a healthy meal, was associated with a higher diet quality, including higher intakes of fruit, vegetables, salads, and fruit juices [73,74,75]. These findings suggest that education in food preparation and time management skills might be needed to alleviate barriers to healthy eating especially among young African Americans.

Additionally, there seems to be a disconnect in the current sample, in that participants reported high levels of self-efficacy and social support for healthy eating but their diet quality was still in the category of poor or needs improvement. Previous studies have showed a similar disconnect [73,75]. For example, Pawlak and Colby found that although African Americans reported high self-efficacy to purchase healthy foods, this did not translate to intakes of these foods [73]. Several possible reasons might be responsible for this discrepancy. First, some African Americans may perceive healthy eating as not being aligned with their cultural heritage [76,77]. The emphasis on soul food in African American culture might make engaging in a healthy diet difficult, as these foods are often high in fat and calorie density [68]. Second, studies have shown that while African Americans may be aware of the benefits of eating healthy, there is a general lack of knowledge regarding dietary guidelines [73,75,78]. For example, in a study comparing the knowledge, attitudes, and beliefs towards healthy foods among African American and Hispanic mothers, African American mothers had comparatively lower knowledge scores related to food groups and dietary recommendations [69]. Third, it is also quite possible that participants lacked knowledge of what constitutes “low fat”, “high fat snack”, and “healthy” foods. These concepts are polysemous and therefore could have resulted in some discrepancy in self-efficacy [79].

This study is not without limitations. The cross-sectional design limits the ability to make conclusions regarding causality. In addition, the sample was from a southeastern metropolitan area thus limiting the generalizability of the results. The use of a convenience sample could also have resulted in under or overrepresentation of particular groups of African Americans. Further, self-reported data were used which are subject to bias and both random and systematic error [35]. Despite these limitations, several findings of the study were supported by the results of large, representative national samples. The strength of this study was that it collectively examined demographics, psychosocial factors, and barriers to healthy eating associated with diet quality among an African American population, thus adding to the extant literature.

Future research should continue to focus on the complex factors associated with diet quality among African Americans, especially young African Americans. Future interventions should focus on increasing intake of whole fruit and dairy, while reducing the consumption of empty calories. Education on the Dietary Guidelines and its recommendation should be an integral part of these interventions. Qualitative studies are needed to understand the perception, facilitators, barriers, and the potential disconnection between perceived ability and support, and actual diet quality among adult African Americans.

## Figures and Tables

**Table 1 nutrients-11-00519-t001:** Characteristics of study participants (*n* = 100).

Gender, *n* (%)	Female	70 (70.0)
	Male	30 (30.0)
Age, *n* (%)	18–35	48 (48.0)
	36–50	25 (25.0)
	51–60	22 (22.0)
	>61	5 (5.0)
Income, *n* (%)	<$20,000	38 (38.0)
	$20,000–$35,000	30 (30.0)
	$35,001–50,000	16 (16.0)
	$51,000–75,000	8 (8.0)
	>$75,000	8 (8.0)
Education, *n* (%)	High school	28 (28.0)
	Some college	52 (52.0)
	College graduate	20 (20.0)
HEI, *n* (%) ^a^	Poor	42 (42.0)
	Needs improvement	58 (58.0)
	Good	0 (0)

^a^ HEI measures diet quality or how well a set of food aligns with the Dietary Guidelines for Americans. HEI = healthy eating index. HEI Grading Scale < 51 = poor, HEI 51–80 = needs improvement, HEI > 80 = good.

**Table 2 nutrients-11-00519-t002:** Comparison of total HEI and subcomponents among study participants and national samples.

HEI Component (Maximum Score)	Sample Averages Mean (SD)	National Averages Mean (SE)	*p*-Value
Total HEI-2010 (100)	54.8 (10.9)	54.3 (1.16)	0.52
Total vegetables (5) ^a^	3.3 (1.2)	3.4 (0.11)	0.57
Greens and beans (5) ^a^	3.4 (1.50)	2.8 (0.19)	0.00 *
Total fruit (5) ^a^	3.3 (1.4)	2.6 (0.14)	0.00 *
Whole fruit (5) ^a^	2.8 (1.5)	3.5 (0.20)	0.00 *
Whole grains (10) ^a^	2.8 (2.0)	2.0 (0.13)	0.00 *
Dairy (10) ^a^	4.1 (1.9)	5.7 (0.18)	0.00 *
Total protein foods (5) ^a^	4.5 (.8)	5.0 (0.00)	0.00 *
Seafood and plant protein (5) ^a^	3.2 (1.5)	3.5 (0.19)	0.07
Fatty acid (10) ^a^	6.4 (2.3)	4.2 (0.15)	0.00 *
Sodium (10) ^b^	4.5 (2.5)	4.20 (0.13)	0.15
Refined grains (10) ^b^	8.0 (2.0)	6.3 (0.12)	0.00 *
Empty calories (20) ^b^	8.5 (4.4)	11.0 (0.41)	0.00 *

^a^ To be consumed in adequate amounts; ^b^ To be consumed in moderate amounts. National averages are for adults aged 18–64 years based on data from the National Health and Nutrition Survey, 2007–2008. * *p* < 0.001.

**Table 3 nutrients-11-00519-t003:** Participants’ self-reported barriers to healthy eating (*n* = 100).

Barriers	Scale	*n* (%)
Others discourage me	Never	69 (69.0)
	Rarely	18 (18.0)
	Sometimes	11 (11.0)
	Often	1 (1.0)
	Very Often	1 (1.0)
I do not have time to prepare healthy foods	Never	17 (17.0)
	Rarely	17 (17.0)
	Sometimes	41 (41.0)
	Often	17 (17.0)
	Very often	8 (8.0)
I am too tired to prepare healthy food	Never	23 (23.0)
	Rarely	27 (27.0)
	Sometimes	41 (41.0)
	Often	8 (8.0)
	Very often	0 (0)
Fast food is more convenient than preparing healthy food	Never	10 (10.0)
	Rarely	7 (7.0)
	Sometimes	31 (31.0)
	Often	27 (27.0)
	Very often	25 (25.0)
Healthy food cost too much	Never	20 (20.0)
	Rarely	20 (21.0)
	Sometimes	31 (31.0)
	Often	15 (15.0)
	Very often	14 (14.0)
I don’t like the taste of healthy food	Never	30 (30.0)
	Rarely	26 (26.0)
	Sometimes	31 (31.0)
	Often	4 (4.0)
	Very often	9 (9.0)
I don’t like to eat healthy food	Never	33 (33.0)
	Rarely	32 (32.0)
	Sometimes	27 (27.0)
	Often	3 (3.0)
	Very often	5 (5.0)
I don’t have access to healthy food	Never	44 (44.0)
	Rarely	25 (25.0)
	Sometimes	21 (21.0)
	Often	7 (7.0)
	Very often	4 (4.0)
I don’t know how to prepare low fat or healthy food	Never	37 (37.0)
	Rarely	19 (19.0)
	Sometimes	23 (23.0)
	Often	13 (13.0)
	Very often	8 (8.0)
Total Barriers, mean (SD)		1.4 (.6)

SD = standard deviation. Range (0 = Never, 1 = Rarely, 2 = Sometimes, 3 = Often, 4 = Very often).

**Table 4 nutrients-11-00519-t004:** Participants’ self-reported of self-efficacy toward healthy eating (*n* = 100).

Self-Efficacy	Scale	*n* (%)
Could you eat low fat or healthy food when at a restaurant	I’m sure I couldn’t	4 (4.0)
	Maybe I couldn’t	4 (4.0)
	Maybe I could	41 (41.0)
	I’m sure I could	48 (48.0)
	Does not apply	3 (3.0)
Could you eat low fat or healthy food when at a family/social gathering	I’m sure I couldn’t	10 (10.0)
	Maybe I couldn’t	16 (16.0)
	Maybe I could	37 (37.0)
	I’m sure I could	34 (34.0)
	Does not apply	3 (3.0)
Could you avoid high fat snack foods when you are in a hurry and don’t have time to prepare a meal	I’m sure I couldn’t	5 (5.0)
	Maybe I couldn’t	12 (12.0)
	Maybe I could	54 (54.0)
	I’m sure I could	28 (28.0)
	Does not apply	1 (1.0)
Could you eat 5 fruits and vegetables a day	I’m sure I couldn’t	6 (6.0)
	Maybe I couldn’t	11 (11.0)
	Maybe I could	34 (34.0)
	I’m sure I could	47 (47.0)
	Does not apply	2 (2.0)
Could you request low fat or healthy food options from a grocery store manager	I’m sure I couldn’t	6 (6.0)
	Maybe I couldn’t	9 (9.0)
	Maybe I could	39 (39.0)
	I’m sure I could	43 (43.0)
	Does not apply	3 (3.0)
Could you request low fat or healthy food options when at a restaurant	I’m sure I couldn’t	6 (6.0)
	Maybe I couldn’t	7 (7.0)
	Maybe I could	35 (35.0)
	I’m sure I could	50 (50.0)
	Does not apply	2 (2.0)
Total self-efficacy, mean (SD)		3.1 (0.7)

SD = standard deviation. Range (0 = Refused, 1 = I’m sure I couldn’t, 2 = Maybe I couldn’t, 3 = Maybe I could, 4 = I’m sure I could).

**Table 5 nutrients-11-00519-t005:** Participants’ self-reported social support for healthy eating (*n* = 100).

Social Support	Scale	*n* (%)
If you had someone like a friend or family member to eat healthier with, chances are that you would eat healthier more often,	Strongly agree	39 (39.0)
	Agree	52 (52.0)
	Disagree	8 (8.0)
	Strongly disagree	1 (1.0)
Your friends encourage you to eat healthy	Strongly agree	18 (18.0)
	Agree	47 (47.0)
	Disagree	28 (28.0)
	Strongly disagree	7 (7.0)
You have at least 1 friend who would commit to eat healthy with you	Strongly agree	27 (27.0)
	Agree	47 (45.0)
	Disagree	22 (22.0)
	Strongly disagree	4 (4.0)
Relatives encourage you to eat healthy	Strongly agree	25 (25.0)
	Agree	50 (50.0)
	Disagree	20 (20.0)
	Strongly disagree	5 (5.0)
You have at least 1 relative who would commit to eat healthy with you	Strongly agree	28 (28.0)
	Agree	48 (48.0)
	Disagree	17 (17.0)
	Strongly disagree	6 (6.0)
Total social support, mean (SD)		2.0 (0.6)

SD = standard deviation. Range (0 = Strongly disagree, 1 = Disagree, 2 = Agree, 3 = Strongly agree).

**Table 6 nutrients-11-00519-t006:** Bivariate correlations between variables of interest and HEI scores (*n* = 100).

Variables	HEI	Barriers	Social Support	Self-Efficacy	Education	Income	Age	Sex
HEI	-	−0.21 *	0.21 *	0.22 *	0.22 *	0.13	0.12	0.05
Barriers		-	−0.30 **	−0.09	−0.12	−0.07	−0.15	−0.07
Social support			-	0.34 ***	0.22 *	0.17	−0.05	−0.01
Self-efficacy				-	0.24 *	0.15	−0.07	−0.02
Education					-	0.23 *	−0.31 **	0.07
Income						-	0.16	−0.11
Age							-	0.01
Sex								-

HEI = healthy eating index. * *p <* 0.05, ** *p <* 0.01, *** *p* < 0.001

**Table 7 nutrients-11-00519-t007:** Linear regressions predicting HEI scores (*n* = 100).

Final Model	Est	SE	CI	*t*	*p*
Age	−0.26	0.16	−0.58–0.07	−1.56	0.12
Sex	1.15	2.19	−3.15–5.44	0.52	0.60
Income	0.10	0.43	−0.75–0.95	0.23	0.82
Education	1.69	0.85	0.03–3.36	1.99	0.05
Self-efficacy	2.04	1.43	−0.75–4.84	1.43	0.16
Social support	0.09	1.73	−3.30–3.49	0.05	0.96
Barriers	−12.13	4.45	−20.85–−3.41	−2.73	0.01 *
Age × Barriers	0.25	0.10	0.05–0.46	2.42	0.02 *

Est = estimates, SE = standard error, CI = confidence interval; * *p <* 0.05.

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
