# Peer review of "An Examination of Demographic and Psychosocial Factors, Barriers to Healthy Eating, and Diet Quality Among African American Adults"

_nutrients, 2019, doi:10.3390/nu11030519_

Reviewer 1 Report

Old title: An examination of demographic, psychosocial, and environmental factors of diet quality in African Americans highlights the need to focus on young African American adults

New title:

An examination of demographic and psychosocial factors, barriers to healthy eating, and diet quality in

among African American Adults

Please check all words in the title (see highlighted above).

The authors revised the manuscript per prior reviews and made significant changes for the entire manuscript. The following are suggestions for presentation of the key findings and related discussions.

1.      The title (see above).

2.      Abstract:

- Lines 29 and 359 should be consistent on the nested linear regression.

- Line 35: poor diet quality: please define this as HEI score of<51? as poor or what exact value, or did the authors meant lower HEI score as poorer diet quality? Once diet quality is defined with HEI score, the results should stay with HEI score in the abstract for scientific specificity.

- Line 37 Americans: it would be more fit to say AA for the conclusion of the study to be based on findings of AA, not all Americans.  

- Please add HEI items (whole fruits, dairy, protein) that are lower scored items for AA in this study compared to the national average referenced.

3.      Introduction:

-Line 42: African American (AA) is abbreviated in the abstract but not in the text. Please be consistent on the use of AA.

4.    Results:

       - Table 1. Footnote: Lines 407-408: a) Please abbreviate HEI in Line 407, then do not spell out HEI in Line 408; b) HEI >80 (correct 81) = Good.

       - Lines 750-752: Please include the p values for those with no barriers and barriers of 2. Table 7 presented a combined p values for the interaction terms of age and barriers, so these p values in the text should be presented for future studies (or a Figure would delineate these associations clearly.

5.    Discussion: Lines 972-973: The authors presented the HEI components that are higher but failed to point out the HEI components that are lower scored or more deficient for AA from this study, including whole fruit, dairy, and protein.  Please include these lower scored items in the Discussion and Abstract for recommendations to AA, and possibly associate these lower scored items in the context of HEI and/or chronic health conditions.

Author Response

Comments and Suggestions for Authors

Old title: An examination of demographic, psychosocial, and environmental factors of diet quality in African Americans highlights the need to focus on young African American adults

New title:

An examination of demographic and psychosocial factors, barriers to healthy eating, and diet quality in

among African American Adults

Please check all words in the title (see highlighted above).

The authors revised the manuscript per prior reviews and made significant changes for the entire manuscript. The following are suggestions for presentation of the key findings and related discussions.

1.     The title (see above).

Response: the word “in” has been removed. 

2.      Abstract:

- Lines 29 and 359 should be consistent on the nested linear regression.

Response: The word “nested” has been added to be consistent.

- Line 35: poor diet quality: please define this as HEI score of<51? as poor or what exact value, or did the authors meant lower HEI score as poorer diet quality? Once diet quality is defined with HEI score, the results should stay with HEI score in the abstract for scientific specificity.

Response: The authors agree that both clarity and scientific specificity are needed and have changed the sentence to read:

“Barriers to healthy eating (b = -12.13, p =0.01) and the interaction between age and barriers (b = .25, p = 0.02) were most strongly associated with poor lower HEI scores. Younger AA’s African Americans with the highest barriers to healthy eating had the lowest HEI scores.”

 - Line 37 Americans: it would be more fit to say AA for the conclusion of the study to be based on findings of AA, not all Americans.

 Response: The statement is referring to the Dietary Guidelines for Americans (DGA) which are the federally set guidelines for all Americans not just African Americans. We see how this could be misinterpreted. We added for African Americans at the end of the sentence in the hope that this will show the distinction.

 - Please add HEI items (whole fruits, dairy, protein) that are lower scored items for AA in this study compared to the national average referenced.

Response: We added whole fruits, dairy and total protein foods.

 3.      Introduction:

-Line 42: African American (AA) is abbreviated in the abstract but not in the text. Please be consistent on the use of AA.

 Response: We replaced AA with African Americans in the abstract so that we are consistent with the text.

 4.    Results:

- Table 1. Footnote: Lines 407-408: a) Please abbreviate HEI in Line 407, then do not spell out HEI in Line 408; b) HEI >80 (correct 81) = Good.

Response: The requested changes have been made to the footnote of Table 1.

- Lines 750-752: Please include the p values for those with no barriers and barriers of 2. Table 7 presented a combined p values for the interaction terms of age and barriers, so these p values in the text should be presented for future studies (or a Figure would delineate these associations clearly. 

Response: We now realize that our presentation was not adequately clear. An interaction between two non-discrete predictors is often difficult to directly interpret as it implies that the effect of one variable on an outcome is changed by the specific level of the other variable. As such, the text in question was merely meant to describe and clarify for the reader how the effect of age varies with the number of barriers. Hence, the text in question is just a description of the interaction between age and barriers and should be viewed as explication of the results in Table 7 and not an additional result that cannot be directly gleaned from Table 7. As you carefully pointed out, the p value for those with no barriers and barriers is the corresponding p-value in Table 7. This p-value clearly describes the level of statistical significance for the interaction term. Therefore, we feel that there was no need to add additional p-values.

We also made the following changes in the text for further clarity:

There was a significant interaction between age and barriers to healthy eating (b = .25, p = 0.01) predicting HEI scores. For example, the results of the interaction in Table 7 imply that for those with no barriers to healthy eating, a decrease in age by 1 year was associated with an increase in average HEI scores of 0.26. For those with 1 barrier to healthy eating age had no impact on HEI scores, and for those with 2 barriers to healthy eating, a decrease in age by 1 year was associated with a decrease in average HEI scores of 0.25. See Table 7 for the results of model 4.

5.    Discussion: Lines 972-973: The authors presented the HEI components that are higher but failed to point out the HEI components that are lower scored or more deficient for AA from this study, including whole fruit, dairy, and protein.  Please include these lower scored items in the Discussion and Abstract for recommendations to AA, and possibly associate these lower scored items in the context of HEI and/or chronic health conditions.

Response:  We added the following paragraph in response to the reviewer’s comment.

Previous research showed African Americans consumed fewer servings of milk and dairy products and therefore had low calcium intake [58,59]. The role of calcium in the reduction of risk for obesity, hypertension, cardiovascular disease, insulin resistance syndrome, and some types of cancers is well established in the literature [59–63]. African Americans and other ethnic populations tend to be lactose intolerant, therefore making it difficult to consume dairy products [64]. Bronner et al suggested the importance of nutrition education interventions that address the cultural, community, and environmental barriers associated with reduced dairy intake in African Americans [61]. Participants consumed less whole fruit but more total fruit. Past research also revealed that African Americans consumed less whole fruit compared to other ethnic groups [55]. It is important to develop strategies to increase the consumption of whole fruit among African Americans as some studies show that 100% fruit juice accounts for approximately one-third of total fruit consumption [65]. Fruit juices are generally devoid of fiber and are a source of liquid calories [66].

Reviewer 2 Report

Reviewer’s Comments

Summary 

The revised manuscript, titled “An examination of demographic and psychosocial factors, barriers to healthy eating, and diet quality among African American Adults” is much improved from the original submission. The authors significantly revised the introduction section of the paper and included more in-depth description of the health and nutrition status of the target population of the study. The methods section has been also revised to provide readers with greater details about the measures and variables used in the analyses. I would like to thank the authors for addressing the main issues that were raised in the review. I have a couple of minor issues to be addressed by the author.  

Remaining issues to be addressed

Lines 1125                extra space, please remove

Lines 1133-1145      Another reason the authors might discuss is related to the poor nutrition knowledge associated with specific foods that are “healthy.” Participants’ self-efficacy might be high because the think they purchase relatively healthy foods. Previous research has reported on misconceptions related to whole bread versus wheat bread, juice products etc.

Author Response

Summary 

The revised manuscript, titled “An examination of demographic and psychosocial factors, barriers to healthy eating, and diet quality among African American Adults” is much improved from the original submission. The authors significantly revised the introduction section of the paper and included more in-depth description of the health and nutrition status of the target population of the study. The methods section has been also revised to provide readers with greater details about the measures and variables used in the analyses. I would like to thank the authors for addressing the main issues that were raised in the review. I have a couple of minor issues to be addressed by the author.  

Remaining issues to be addressed

Lines 1125                extra space, please remove

Response: The extra space has been removed.

Lines 1133-1145      Another reason the authors might discuss is related to the poor nutrition knowledge associated with specific foods that are “healthy.” Participants’ self-efficacy might be high because they think they purchase relatively healthy foods. Previous research has reported on misconceptions related to whole bread versus wheat bread, juice products etc.

Response: We agree with the suggestion regarding poor nutrition knowledge being associated with specific foods that are “healthy”. We have added the following statement.

It is also quite possible that participants lacked knowledge of what constitutes “low fat”, “high fat snack”, and “healthy” foods. These concepts are polysemous and therefore could have resulted in some discrepancy in self-efficacy [79]

This manuscript is a resubmission of an earlier submission. The following is a list of the peer review reports and author responses from that submission.

Round  1

Reviewer 1 Report

The authors presented a study about healthy eating for diet quality in African American adults. The following are suggestions for presentation of the key findings and related discussions.

1.      The title of this study should be reworded for a concise association of the variables of interests. For clarify, the title can be reworded considering African Americans are mentioned twice, diet quality meant healthy eating index, demographic factors only presented age as a significant factor, none of the environmental factors are clearly presented in the abstract for significance.

2.      Introduction: There is a major disconnect from the sections of the introduction to the significance for so what kind of intervention can be meaningful for future or so what domains of diet for health eating is  major area for improvement in the population of interest.

3.      In the Method’s section, please clarify the conceptual domains on the social versus environmental factors.

4.      In the results section, please add itemized HEI scores on which items the cases fall short on. Just listing total HEI score does not help the scientific community to understand the deficiency domains for the AA populations on what they eat more or less.

5.      For Tables, please include only the significant findings for the regression. Many insignificant variables are included that present confusion in the fidnings. The major limitation of this study as presented, are weak with few variables of significance. More advanced statistics such as machine learning based analytics can be considered to yield more powerful results.

6.      In the Discussion, the discussions lack tight connections to the key significant findings for scientific discovery. Please present the most significant key findings in relation to healthy eating for AA to improve their dietary profile in relation to the health outcomes. Total HEI score may be detailed enough to provide insights for improvement of healthy eating lifestyles.

Reviewer 2 Report

A brief summary 

The manuscript, titled “An examination of demographic, psychosocial, and environmental factors of diet quality in African Americans highlights the need to focus on young African American adults,” aims to examine the associations between social, physical and environmental factors and diet quality among African American adults. This is a cross-sectional study with a sample of 102 participants. While the study is related to an important topic of health disparities among African Americans in the U.S., the aims of the paper have been examined in a number of previous research studies. Thus, the findings of the manuscript, in its current form, do not add innovative contributions to the current literature related to diet quality and health disparities among African American adults. 

Broad comments

The strength of the paper is that the authors have a variety of personal, social, and environmental data available for analyses, including self-efficacy, social support for healthy eating, and barriers to healthy eating. Another strength of the paper is that the authors were able to utilize data from 24-hour dietary recalls and calculate overall diet quality from the available data. This is unique as many studies rely on selected dietary variables without examining overall diet quality. However, several weaknesses must be also noted. While the paper attempts to address racial health disparities by examining correlates of diet quality in a sample of adult African American adults, the work presented in the paper is lacking unique hypothesis and/or research questions and related depth of data that would help explain the development or maintenance of chronic disease risk among African American individuals. The introduction section does not outline clear/strong links (with appropriate references) between the known personal, social and environmental factors and diet-related outcomes in the target sample and thus, does not present a convincing argument for the need to conduct the current study. If the authors can present a more in-depth case and make significant revisions in the introduction section, it would help strengthen the paper. In addition, the manuscript can be much improved by revisiting the writing style and grammar. There are several places in the text where the meaning of the statements is vague and should to be clarified.  Specific examples and suggestions are listed below. 

Specific comments 

Introduction 

The introduction section provides less than a strong argument for the purpose of the study. Some of the statements are unclear and should be revised. 

Lines 48-52: … “and higher incidences of chronic diseases (10-12).” Higher than who? Please clarify?

Lines 53-61    In this paragraph, a better synthesis of previous research is warranted to highlight why the current study is needed. It is not clear what was found in previous research about barriers, self-efficacy and social support in previous studies and what type of research has found it (i.e., qualitative research?).  

Methods 

Lines 114-116 Please correct the references and change it to a numbered reference for the FFQ. Also, add a correct reference for the Block FFQ. It is unclear which measure was used in the study. 

Lines 115 How many items were included in the FFQ? The measure needs to be described in a greater detail so other research using the measure know what was done and they can follow the similar procedures. 

Lines 125-131 Please reference papers that describe how HEI-2010 is used and how overall diet quality is derived. The reference provided is from 1995, which does not reflect the description and procedures related to HEI-2010. 

Lines 84 The methods section seems to list the construct assessed in the study, with a description of the construct and a specific measure used to assess it. On line 84, the authors list the REACH 2010 St. Louis Healthy Heart Survey which is a measure, not a construct. Please be consistent with the way you outline the constructs and measures in the methods section.    

Results 

Table 1. The footnote should be revised, there are spelling issue (i.e., “competed” instead of “completed”) and the formatting of the footnote is incorrect. 

Table 2. Title of Table 2. Now reads “nine items that assessed barriers to healthy eating.” Please correct the grammar and revise the title. 

Table 2, Table 3 and Table 4. Please provide range of scores for the items in the Tables 

Discussion 

Lines 212 Please explain what it means “participants had “increased” social support? Compared to what? 

Line 229-231 The authors state that “young adults are expected to be at peak health and performance.” As authors know, a lack of chronic disease does not mean that individuals consume a high quality diet, and such findings would not be detectable in cross-sectional research.     

Lines 281-287 "Author Contributions" section is missing any information